# Apolipoprotein CIII Reduction Protects White Adipose Tissues against Obesity-Induced Inflammation and Insulin Resistance in Mice

**DOI:** 10.3390/ijms23010062

**Published:** 2021-12-22

**Authors:** Patricia Recio-López, Ismael Valladolid-Acebes, Per-Olof Berggren, Lisa Juntti-Berggren

**Affiliations:** The Rolf Luft Research Center for Diabetes and Endocrinology, Karolinska Institutet, Karolinska University Hospital, SE-171 76 Stockholm, Sweden; patricia.recio.lopez@ki.se (P.R.-L.); per-olof.berggren@ki.se (P.-O.B.); lisa.juntti-berggren@ki.se (L.J.-B.)

**Keywords:** apolipoprotein CIII, diet-induced obesity, insulin resistance, white adipose tissue, inflammation, antisense oligonucleotides

## Abstract

Apolipoprotein CIII (apoCIII) is proinflammatory and increases in high-fat diet (HFD)-induced obesity and insulin resistance. We have previously shown that reducing apoCIII improves insulin sensitivity in vivo by complex mechanisms involving liver and brown adipose tissue. In this study the focus was on subcutaneous (SAT) and visceral (VAT) white adipose tissue (WAT). Mice were either given HFD for 14 weeks and directly from start also treated with antisense oligonucleotide (ASO) against apoCIII or given HFD for 10 weeks and HFD+ASO for an additional 14 weeks. Both groups had animals treated with inactive (Scr) ASO as controls and in parallel chow-fed mice were injected with saline. Preventing an increase or lowering apoCIII in the HFD-fed mice decreased adipocytes’ size, reduced expression of inflammatory cytokines and increased expression of genes related to thermogenesis and beiging. Isolated adipocytes from both VAT and SAT from the ASO-treated mice had normal insulin-induced inhibition of lipolysis compared to cells from Scr-treated mice. In conclusion, the HFD-induced metabolic derangements in WATs can be prevented and reversed by lowering apoCIII.

## 1. Introduction

Obesity is defined as abnormal or excessive fat accumulation that poses a health risk [1,2]. The fundamental cause of obesity and overweight is an energy imbalance between calories consumed and calories expended [3]. Hypertriglyceridemia often occurs in obesity and there is a positive correlation between apolipoprotein CIII (apoCIII) and triglycerides [4,5,6]. In recent years apoCIII has become interesting as a co-actor in the development of overweight and insulin resistance [7,8,9]. During conditions of insulin resistance, there is a parallel increase in intra-islet apoCIII associated with local islet inflammation and β-cell dysfunction/apoptosis [10]. It has been shown that both transgenic mice, overexpressing apoCIII, and mice with a complete deficiency of the apolipoprotein have a more pronounced diet-induced obesity (DIO) [5,6,11]. However, decreasing apoCIII has beneficial metabolic effects [9,10,12]. The degree of apoCIII suppression required to induce protective effects against metabolic derangements ranges from 40 to 98%, depending on species, strains, diets and genetic factors [9,10,12,13,14]. When C57BL6/j mice on high-fat diet (HFD) were treated with antisense oligonucleotides (ASOs) against apoCIII, either directly from start or after 10 weeks on the diet, the harmful effects of the HFD were eliminated in both groups [9].

The underlying mechanisms behind these positive results of lower apoCIII levels are complex and involve increased lipase activity and receptor-mediated hepatic uptake of lipids, normalization of insulin sensitivity, increase in hepatic fatty acid oxidation and ketogenesis, while de novo lipogenesis and gluconeogenesis are inhibited [9]. In addition, there are several pieces of evidence for a metabolically more active brown adipose tissue with an increase in expression levels of β3-adrenergic receptor, body temperature, lipoprotein lipase activity and upregulation of thermogenic genes [9]. 

The absence of obesity in ASO-treated mice fed an HFD made us interested in comparing white adipose tissue (WAT) from untreated and treated mice. Both preventing and reversing HFD-induced increase in apoCIII result in morphological and metabolic improvements in visceral (VAT) and subcutaneous (SAT) WATs. These data further support that apoCIII can be a target in the development of new treatments for obesity and its consequences.

## 2. Results

Mice were fed HFD from eight weeks of age. A prevention group was directly from the start of the diet, which lasted for 14 weeks, treated twice per week i.p. with active or inactive (Scr) ASO against apoCIII (Scr 14w and ASO 14w). In a reversibility group, the animals were first fed HFD for 10 weeks and thereafter, still continuing on HFD, treated with ASO or Scr for 14 weeks (HFD+Scr and HFD+ASO). Both studies had control groups on chow diet given saline i.p. (Sal 14w and CD+Sal).

### 2.1. ApoCIII Reduction Improves Metabolic Status in HFD-Fed Mice

At the end of the study, the expression levels of liver *apoCIII* mRNA were analyzed, and this confirmed that all ASO-treated mice had low levels, while the HFD-fed mice without an apoCIII lowering treatment had an upregulation of the gene (Figure 1a). Mice on HFD treated with Scr were obese and had a diabetic phenotype with hyperglycemia, hyperinsulinemia and hyperglucagonemia (Figure 1b–e). As a consequence of obesity, the levels of adiponectin, a marker of insulin sensitivity, were decreased, while leptin, as expected, was increased in the obese mice (Figure 1f,g). Treatment with ASO, in both the prevention and reversibility studies, resulted in body weights (BWs) similar to those of mice in the control groups on chow and a normalization of glucose, insulin, glucagon, adiponectin and leptin levels (Figure 1b–g). There were no differences in plasma low-density lipoprotein- (LDL) and high-density lipoprotein- (HDL) cholesterol levels between control and ASO-treated mice, whereas they were elevated in Scr-treated mice (Figure 1h,i). 

### 2.2. Lowering apoCIII Reduces Inflammation and Size of Adipocytes in WAT

The HFD-induced hypertrophy of adipocytes in SAT and VAT in Scr-treated mice was prevented and reversed by decreasing apoCIII with ASO (Figure 2a–h). Elevated levels of apoCIII are related to inflammation [10,15], and there was an upregulation of the expression of genes for the inflammatory cytokines interleukin-1beta (*IL*-*1β*), interleukin-6 (*IL*-*6*) and tumor necrosis factor-α (*TNF*-*α*) in the obese, compared to ASO-treated and control, mice (Figure 2i–n).

### 2.3. Reduction in apoCIII Increased Expression of Genes Related to Thermogenesis and Improved Function of SAT

There was a downregulation of the thermogenic genes uncoupling protein-1 (*UCP-1*), peroxisome proliferator-activated receptor gamma coactivator-1alpha (*PGC-1α*), PR domain containing 16 (*PRDM16*) and cell death inducing DFFA like effector A (*CIDEA*) in SAT from Scr14w mice (Figure 3a–d). In the ASO14w group, the expression of *PGC-1α* and *PRDM16* in SAT did not differ from control mice, whereas *UCP-1*, *CIDEA* and iodothyronine deiodinase 2 (*DIO2*) were higher (Figure 3a–e). 

Similar, although less pronounced, results were found in SAT from the reversibility study where the obese mice were first on HFD for 10 weeks and thereafter on HFD and ASO for 14 weeks. Reduction in apoCIII also affected genes related to beiging of WAT. *CD137* (tumor necrosis factor receptor superfamily member 9, TNFRSF9), transmembrane protein 26 (*TMEM26*) and T-box protein 1 (*TBX1*) increased in both ASO-treated groups (Figure 3f–h). Isolated adipocytes from SAT from HFD-fed Scr-treated mice had an upregulated expression of adipose triglyceride lipase (*ATGL*) and a suppression of hormone-sensitive lipase (*HSL*) genes (Appendix A). Functional in vitro studies showed that there was increased basal lipolysis but blunted isoproterenol-induced lipolysis in the adipocytes from the Scr-treated mice (Figure 3i,j and Appendix A). ASO treatment normalized the expression levels of *ATGL* and *HSL*, as well as basal and isoproterenol-induced lipolysis (Figure 3i,j and Appendix A). 

### 2.4. Reduced Levels of apoCIII and VAT Function

The gene markers for thermogenesis and beiging, analyzed in SAT (Figure 3a–h), were also determined in VAT. All ASO-treated mice, compared to Scr-treated mice, had an upregulation of these genes (Figure 4a–h). As in SAT, there was a normalization of the expression of the genes for *ATGL* and *HSL* lipases and isoproterenol-induced lipolysis by decreasing apoCIII (Figure 4i,j and Appendix A). 

### 2.5. ApoCIII Reduction during HFD Maintains and Restores Insulin Sensitivity in WAT

As insulin resistance is a major problem in obesity and type 2 diabetes with impaired inhibition of lipolysis, we tested whether lowering of apoCIII improves this parameter in adipocytes from VAT and SAT. We could demonstrate that adipocytes from VAT and SAT from both the prevention group, which had never been exposed to increased levels of apoCIII, and from the reversibility study, where apoCIII was decreased during the last 14 weeks of the 24-week study, revealed a normal insulin-induced inhibition of lipolysis (Figure 5a–d). Conversely, cells from Scr-treated mice were all resistant to insulin (Figure 5a–d).

## 3. Discussion

Increased levels of apoCIII induce hypertriglyceridemia by inhibiting LPL activity and hepatic clearance of circulating lipids by blocking liver lipoprotein receptors [14,16,17,18]. There are subjects with mutations in the gene encoding LPL or genes encoding proteins necessary for LPL function (familial chylomicronemia syndrome (FCS)) [14]. FCS is characterized by severe hypertriglyceridemia. These patients do not respond to conventional lipid lowering treatment, but ASO against apoCIII has been shown to be effective [14]. FCS is a rare disease, while obesity and its complications including type 2 diabetes (T2D) are a major, and unfortunately growing, problem that also affects young people. 

We have previously demonstrated that the metabolic derangements induced by HFD can be prevented if animals directly from the start of the diet are treated with ASO against apoCIII, thus preventing an increase in the apolipoprotein [9]. An even, from our point of view, more interesting finding was that if already obese animals on HFD were put on ASO treatment the metabolic phenotype was restored, despite the animals continuing on HFD [9]. This mimics the clinical situation where people come for treatment when they are already obese and have T2D.

In our previous studies, we investigated the underlying mechanisms behind the beneficial effects of apoCIII lowering treatment in liver, brown adipose tissue (BAT) and islets of Langerhans [9,10]. In this study, we focused on subcutaneous and visceral WAT because it was striking to see that mice on HFD and ASO, who did not eat less or move more than their obese Scr-treated controls, had the same fat mass as control animals on chow. Volanesorsen is a 2′-O(2-methoxyethyl)-modified antisense against apoCIII used in the clinic to treat FCS, and it has also been shown to lower triglycerides, and thereby reduce the risk of acute pancreatitis, in patients with multifactorial chylomicronemia [14,19]. However, there is to our knowledge a lack of data on the effects in WAT of apoCIII lowering with ASO. Furthermore, WAT is highly interesting in the development of low-grade inflammation and insulin resistance associated with the metabolic disturbances seen in obesity and T2D [20]. 

Again, we could show that by inhibiting or reducing the HFD-induced increase in apoCIII, the obesity-related metabolic disorders are prevented or restored. Focusing on WAT there was a hypertrophy of adipocytes in both SAT and VAT from Scr-treated obese mice. The difference in size and that VAT has larger adipocytes than SAT are in line with previous observations [21,22,23]. The adipocytes normalized their size in both tissues after ASO treatment. There are suggestions that the size difference is a determining factor for a pathogenic or protective role associated with VAT and SAT, respectively [21,22,23]. It has been demonstrated that obese subjects without metabolic complications have, in contrast to obese with metabolic impairments, smaller and more numerous adipocytes in WAT accompanied with a lower degree of inflammation [23]. In a population of 60 patients with obesity, where two-thirds of them met the criteria for the metabolic syndrome, large adipocytes from SAT were associated with a worse gluco-metabolic profile than large adipocytes from VAT [22]. As a possible explanation for this difference, it was suggested that because adipocytes from SAT show a lower expandability which limits the ability of lipid storage, lipid accumulation occurs in other tissues (hepatocytes, myocytes, β-cells), which causes impaired metabolism such as insulin resistance [22]. In a large population study, it was concluded that both SAT and VAT are associated with glucose levels and insulin resistance with a stronger association for VAT [24].

The expression levels of genes for *IL-**β*, *IL-6* and *TNF-**α* were increased in obese mice in both the prevention and the reversibility studies, indicating an ongoing inflammatory process in WAT. When the proinflammatory apoCIII was lowered by ASO, the cytokines decreased. There was also an enhanced expression of genes related to thermogenesis and beiging in the ASO-treated mice, as further evidence of improved metabolism. 

Lipolysis is inhibited by insulin and stimulated by catecholamines, and functional studies in isolated adipocytes from SAT and VAT, from all experimental groups, revealed that lowering apoCIII normalizes these responses. This clearly shows that there is a restored insulin sensitivity in WAT, as previously shown in liver and BAT, which is a key factor for an efficient glucose and lipid metabolism [9].

There are limitations in this study. One is that when we fractionated WATs and isolated adipocytes from the floating fraction, from lean mice this fraction consists mainly of adipocytes from the stromal vascular portion. However, the floating fraction from obese mice is also known to contain immune cells [25,26]. Hence, functional changes in isolated adipocytes from obese mice could be, at least in part, driven by a different tissue composition/architecture.

Another limitation is that these studies have only been done in mice so far and have to be proven in human adipose tissues before broader conclusions can be made. 

In summary, we show that lowering apoCIII reduces the expression of inflammatory cytokines and counteracts the negative metabolic effects of HFD-induced obesity in both SAT and VAT and, together with our previous results in liver and BAT, we entertain the possibility that this can be implemented as a treatment strategy for obesity and insulin resistance.

## 4. Materials and Methods

### 4.1. Mice and Diets

Male C57BL6/j (B6) mice were purchased from Charles River at the age of six weeks. Mice, housed 5 animals per cage, were acclimated to our animal facilities for two weeks from arrival in a temperature- and humidity-controlled room with 12 h light/dark cycles and ad libitum access to water and food (R70, Lantmännen, Stockholm, Sweden). At the age of eight weeks, normoglycemic mice with similar body weight were assigned to be either fed a high-fat diet (HFD; 60% kcal from fat, Open Source Diets D12492, Research Diets, New Brunswick, NJ, USA) or kept on standard chow diet (R70, Lantmännen, Sweden). The duration of the diet intervention varied depending on the experimental design, from 14 to 24 weeks. Animal care and experiments were carried out according to the Animal Experiment Ethics Committee at Karolinska Institutet (Ethical permit number 19462/2017, date of approval, 08-February-2018).

### 4.2. Antisense Oligonucleotides (ASOs)

Chimeric 20-mer phosphorothioate ASOs containing 2′-O-methoxyethyl (2′MOE) groups at positions 1–5 and 16–20 were provided by Ionis Pharmaceuticals, Inc. The sequence of the active antisense (ASO) against apoCIII gene (ION-353982) was 5′-GAGAATATACTTTCCCCTTA-3′ and that of the inactive or scrambled (Scr) (ION-141923) was 5′-CCTTCCCTGAAGGTTCCTCC-3′, as previously described [9]. Scr and active ASO were developed and tested for specificity and toxicity by the IONIS Pharmaceutical Company, as previously described [13]. 

### 4.3. Experimental Design 

All mice were eight weeks old at start. We performed the following studies: (1) prevention study where mice were fed HFD for 14 weeks and simultaneously given ASO (ASO14w) or Scr (Scr14w) intraperitoneally (i.p.) twice per week at a dose of 25 mg/kg; (2) reversibility study where mice were first fed HFD for 10 weeks and thereafter treated i.p. with either ASO (HFD+ASO) or Scr (HFD+Scr) twice per week (25 mg/kg) for 14 additional weeks, still continuing on HFD. Both studies had control groups with mice on chow diet given equivalent volumes of saline i.p. (Sal14w and CD+Sal). 

### 4.4. Hematoxylin and Eosin Staining in SAT and VAT 

Paraformaldehyde-fixed SAT and VAT were sectioned into 20 μm thick sections with a cryostat (Microm HM500M/Cryostar NX70; Thermo Scientific, Uppsala, Sweden) and collected onto SuperFrost Plus microscope slides (VWR International AB, Stockholm, Sweden). The sections were rinsed in hematoxylin (Sigma-Aldrich, Stockholm, Sweden; 50% vol/vol diluted in distilled water) for 6 min, washed with distilled water and stained with eosin (Histolab, Askim, Sweden) for 30 s. After washing, sections were dried and mounted with VectaMount permanent mounting medium (Vector Laboratories, CA, USA). All procedures were performed in a cold room at 4 °C. Once the mounting medium was solidified, the preparations were brought up to room temperature and imaged under a 20× magnification objective using an optical microscope (Leica, Wetzlar, Germany). The area of the adipocytes was analyzed using the ImageJ program (National Institutes of Health, Bethesda, MD, USA).

### 4.5. Functional Studies in Isolated Adipocytes from SAT and VAT

Mature adipocytes were isolated from SAT and VAT as previously described [27,28,29]. To test catecholamine-stimulated lipolysis, isolated adipocytes were incubated for 90 min with 1 U/mL of adenosine deaminase (Merck, Stockholm, Sweden) in the absence (basal lipolysis) or presence of 10-fold increasing concentrations of isoproterenol (Merck), ranging from 10^−8^ to 10^−5^ mol/L. Lipolysis was quantified as liberation of glycerol into the medium, determined by the GPO-Trinder method (Sigma-Aldrich, Stockholm, Sweden). To measure the effect of insulin on lipolysis, isolated adipocytes were incubated with 0, 0.3, 0.6, 1.5 or 4.75 nmol/L insulin (Merck, Stockholm, Sweden) prior to the addition of isoproterenol (10^−7^ nmol/L) and incubated for 90 min.

### 4.6. Blood Glucose and Plasma Biochemistry 

Blood glucose concentrations were measured prior to euthanasia with Accu-Chek Aviva monitoring system (F. Hoffmann-La Roche, Basel, Switzerland). For biochemical determinations in plasma, blood samples from non-fasted mice were obtained at the end of the experiments, collected in Microvette CB 300 K2 EDTA tubes (SARSTEDT AG & Co. KG, Helsingborg, Sweden) and kept on ice. Thereafter, blood samples were centrifuged at 2500× *g* for 15 min at 4 °C, plasma collected and preserved at −80 °C until use. Non-fasting plasma insulin, leptin and adiponectin were measured using mouse-specific ELISA kits (Phoenix Pharmaceuticals, CA, USA for leptin; Crystal Chem, Zaandam, Netherlands, for insulin and adiponectin). LDL and HDL cholesterol were determined by the Trinder reaction using colorimetric assays based on a modified polyvinyl sulfonic acid (PVS) and polyethylene glycol methyl ether (PEGME) coupled classic precipitation method (Crystal Chem, Zaandam, The Netherlands). For glucagon determination, blood samples were collected in a separate set of Microvette CB 300 K2 EDTA tubes (SARSTEDT AG & Co. KG, Helsingborg, Sweden), supplemented with aprotinin (500 KIU/mL) and kept on ice. Immediately after, blood samples were centrifuged as specified above and obtained plasma preserved at −80 °C until use. Plasma glucagon levels were determined using a mouse-specific glucagon ELISA kit (Crystal Chem, Zaandam, The Netherlands). 

### 4.7. RNA Isolation and Quantitative Real-Time Polymerase Chain Reaction (qRT-PCR) 

SAT and VAT tissues were quickly dissected out and samples preserved at −80 °C until use. Total RNA was isolated using the RNeasy Lipid Tissue Mini Kit according to the manufacturer’s protocol (Qiagen, Stockholm, Sweden). Total RNA concentrations were determined using a NanoPhotometer P330 (IMPLEN, Sweden). Four hundred nanograms of total RNA was used for cDNA preparation using the High Capacity cDNA Reverse Transcription kit (Thermo Scientific, Uppsala, Sweden). qRT-PCR was performed in a QuantStudio 5 PCR system thermal cycler (Thermo Scientific, Uppsala, Sweden) with Power-Up SYBR green PCR master mix (Thermo Scientific, Uppsala, Sweden). As control genes for liver apoCIII gene expression we used *β*-*actin*, TATA-binding protein (*TBP*) and hypoxanthine phosphoribosyltransferase 1 (*HPRT*). As control genes for SAT and VAT target transcripts we used TATA-binding protein (*TBP*) and peptidylprolyl isomerase A (*PPIA*). Analysis of gene expression was done with the ΔΔCt method. The relative transcript levels of each target gene were normalized to each control gene and to the geometric mean of the cycle threshold (Ct) of all control genes used for qRT-PCR analyses in SAT and VAT. The results are expressed as mRNA levels relative to saline-treated groups fed with chow (Sal14w for the prevention study or CD+Sal for the reversibility study). Primer sequences are available in Table 1.

### 4.8. Statistical Analysis 

All statistical analyzes were performed using GraphPad Prism 5.0. Nonparametric two-tailed Mann–Whitney U *t*-test was used for comparison of two groups. 2-ANOVA followed by Bonferroni’s post hoc test was used for the analyses of dose–response curves. Statistical significance was defined as *p* < 0.05.

## Figures and Tables

**Figure 1 ijms-23-00062-f001:**
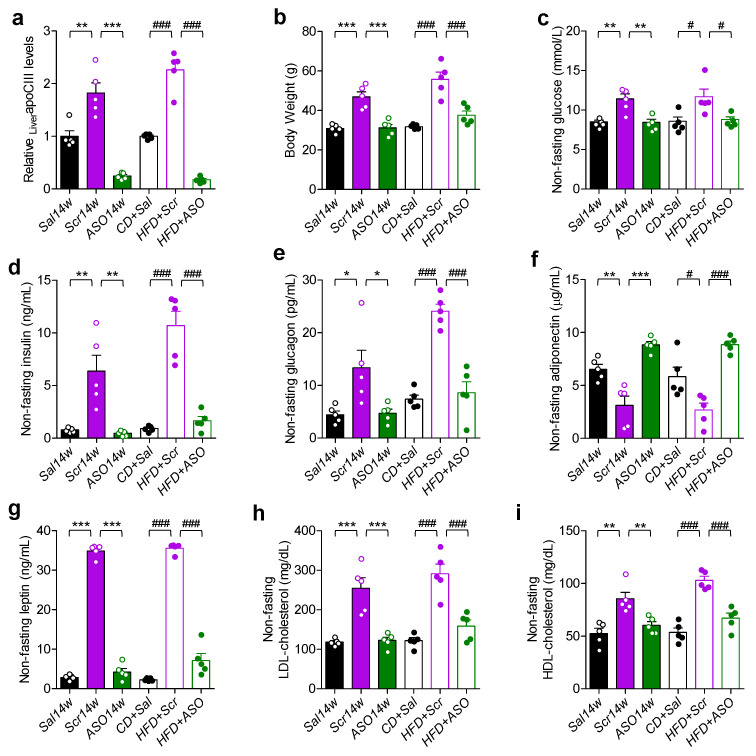
Reducing apoCIII improves metabolic status. (**a**) Liver apoCIII mRNA levels analyzed by qRT-PCR at the end of the prevention (filled bars) and reversibility (empty bars) studies. (**b**) Body weight. (**c**–**i**) Non-fasting (**c**) glucose, (**d**) insulin, (**e**) glucagon, (**f**) leptin, (**g**) adiponectin, (**h**) LDL-cholesterol and (**i**) HDL-cholesterol measured at the end of the prevention (filled bars) and reversibility (empty bars) studies. Data are expressed as mean ± SEM of *n* = 5 individual mice per experimental group and controls. * *p* < 0.05; ** *p* < 0.01; *** *p* < 0.001 for differences in the prevention study. # *p* < 0.05; ### *p* < 0.001 for differences in the reversibility study.

**Figure 2 ijms-23-00062-f002:**
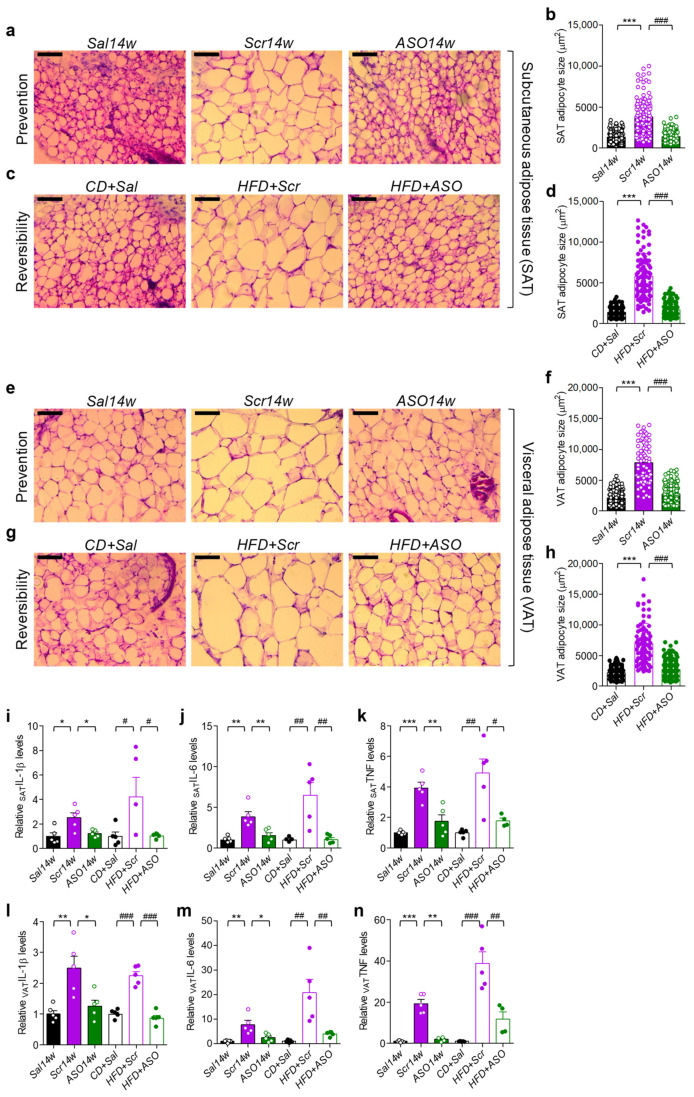
Effect of apoCIII reduction on size of adipocytes and inflammatory cytokines in SAT and VAT. (**a**–**h**) Representative microphotographs and quantitative analyses of hematoxylin and eosin–stained cryosections from SAT from the (**a**,**b**) prevention and (**c**,**d**) reversibility studies and from VAT (**e**,**f**) and (**g**,**h**). Scale bar located in the top left corner of every image is 100 μm. (**i**−**n**) mRNA levels of inflammatory cytokines in SAT and VAT measured by qRT-PCR. SAT (**i**) *IL*-*1**β*, (**j**) *IL*-*6* and (**k**) *TNF*-*α* and VAT (**l**) *IL*-*1**β*, (**m**) *IL*-*6* and (**n**) *TNF*-*α*. Filled bars show results from the prevention and empty bars from the reversibility studies. For the quantitative analyses, data are expressed as mean ± SD of *n* = 3 individual mice per experimental group and controls; for the qRT-PCR experiments, data are expressed as mean ± SEM of *n* = 4–5 individual mice per experimental group and controls. * *p* < 0.05; ** *p* < 0.01; *** *p* < 0.001 for differences in the prevention study. # *p* < 0.05; ## *p* < 0.01; ### *p* < 0.001 for differences in the reversibility study.

**Figure 3 ijms-23-00062-f003:**
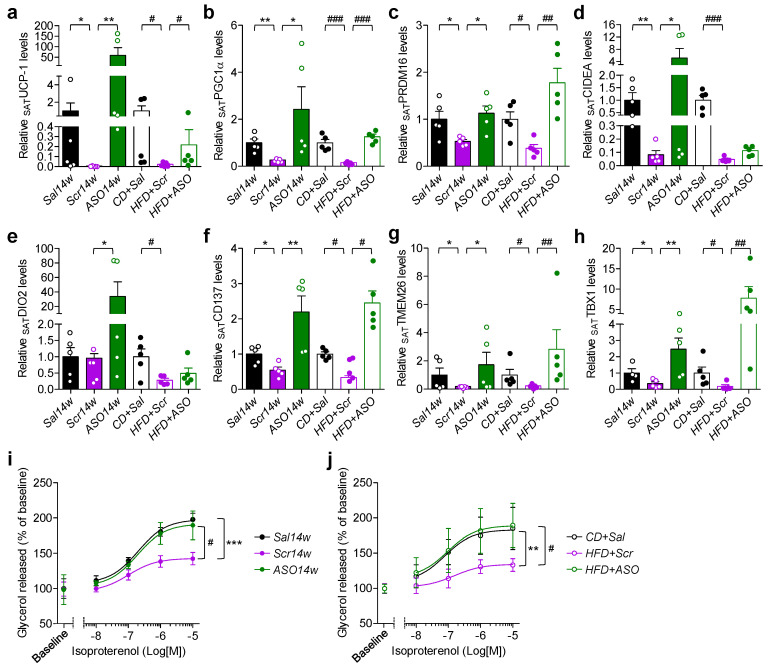
Increase in thermogenic and beiging gene markers by apoCIII reduction improves catecholamine-induced lipolysis in SAT. (**a**–**h**) Expression of (**a**) *UCP*-*1*, (**b**) *PGC1**α*, (**c**) *PRDM16,* (**d**) *CIDEA*, (**e**) *CD137*, (**d**) *TMEM26* and (**d**) *TBX1* mRNA in SAT from the prevention (filled bars) and reversibility (empty bars) studies. (**i**,**j**) Concentration–response curves of isoproterenol-induced lipolysis in isolated adipocytes from SAT from (**i**) the prevention and (**j**) reversibility studies. Data expressed as mean ± SEM of *n* = 4–5 individual mice per experimental group and controls. * *p* < 0.05; ** *p* < 0.01; *** *p* < 0.001 for differences in the prevention study. # *p* < 0.05; ## *p* < 0.001; ### *p* < 0.001 for differences in the reversibility study.

**Figure 4 ijms-23-00062-f004:**
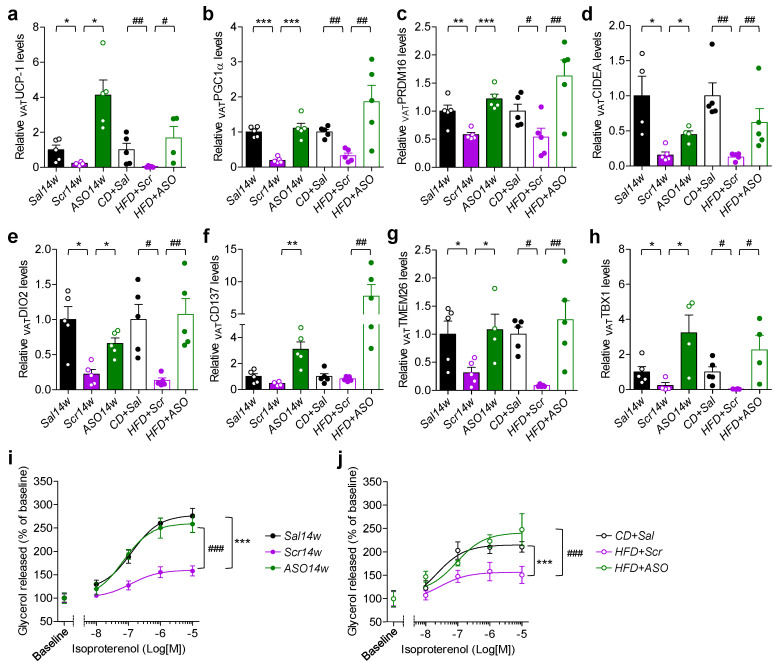
Increase in thermogenic and beiging gene markers by apoCIII reduction improves catecholamine-induced lipolysis in VAT. (**a**–**h**) Expression of (**a**) *UCP*-*1*, (**b**) *PGC1**α*, (**c**) *PRDM16,* (**d**) *CIDEA*, (**e**) *CD137*, (**d**) *TMEM26* and (**d**) *TBX1* mRNA in the prevention (filled bars) and reversibility (empty bars) studies. (**i**,**j**) Concentration–response curves of isoproterenol-induced lipolysis in isolated adipocytes from VAT from (**i**) the prevention and (**j**) reversibility studies. Data expressed as mean ± SEM of *n* = 4–5 individual mice per experimental group and controls. * *p* < 0.05; ** *p* < 0.01; *** *p* < 0.001 for differences in the prevention study. # *p* < 0.05; ## *p* < 0.001; ### *p* < 0.001 for differences in the reversibility study.

**Figure 5 ijms-23-00062-f005:**
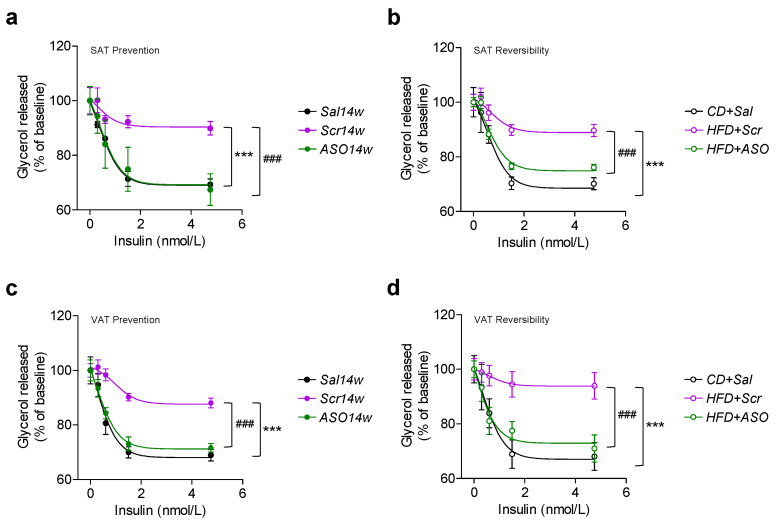
ApoCIII reduction improves WAT insulin sensitivity. (**a**–**d**) Concentration-dependent inhibition of isoproterenol-induced lipolysis by insulin in isolated adipocytes from (**a**,**b**) SAT from mice in (**a**) the prevention and (**b**) the reversibility studies and (**c**,**d**) VAT from mice in (**c**) the prevention and (**d**) the reversibility studies. Data are expressed as mean ± SEM of *n* = 5 individual mice per experimental group and controls. *** *p* < 0.001 for differences in the prevention study. ### *p* < 0.001 for differences in the reversibility study.

**Table 1 ijms-23-00062-t001:** Primers used for gene expression by qRT-PCR. Sequences of each primer pair (forward and reverse) from 5′ to 3′.

Gene ID	Primer Strand	Primer Sequence (5′→3′)	PCR Size (*bp*)
*apoCIII*	Forward	CGCTAAGTAGCGTGCAGGA	68
	Reverse	TCTGAAGTGATTGTCCATCCAG	
*ATGL*	Forward	GGTCCTCCGAGAGATGTGC	75
	Reverse	TGGTTCAGTAGGCCATTCCTC	
*CD137*	Forward	GTCGACCCTGGACGAACTGCTCT	134
	Reverse	CCTCTGGAGTCACACAGAAATGGTGGTA	
*CIDEA*	Forward	ATCACAACTGGCCTGGTTACG	136
	Reverse	TACTACCCGGTGTCCATTTCT	
*DIO2*	Forward	TTGGGGTAGGGAATGTTGGC	99
	Reverse	TCCGTTTCCTCTTTCCGGTG	
*HPRT*	Forward	CAGTCCCAGCGTCGTGATTA	167
	Reverse	GGCCTCCCATCTCCTTCATG	
*HSL*	Forward	TCCTCAGAGACCTCCGACTG	135
	Reverse	ACACACTCCTGCGCATAGAC	
*IL-1β*	Forward	TGGACCTTCCAGGATGAGGACA	148
	Reverse	GTTCATCTCGGAGCCTGTAGTG	
*IL-6*	Forward	TACCACTTCACAAGTCGGAGGC	116
	Reverse	CTGCAAGTGCATCATCGTTGTTC	
*PGC1α*	Forward	CCCTGCCATTGTTAAGACC	161
	Reverse	TGCTGCTGTTCCTGTTTTC	
*PPIA*	Forward	GGGTTCCTCCTTTCACAGAA	145
	Reverse	GATGCCAGGACCTGTATGCT	
*PRDM16*	Forward	CAGCACGGTGAAGCCATTC	87
	Reverse	GCGTGCATCCGCTTGTG	
*TBP*	Forward	TGCTGTTGGTGATTGTTGGT	97
	Reverse	CTGGCTTGTGTGGGAAAGAT	
*Tbx1*	Forward	GGCAGGCAGACGAATGTTC	102
	Reverse	TTGTCATCTACGGGCACAAAG	
*TMEM26*	Forward	GAAACCAGTATTGCAGCACCCAAT	205
	Reverse	AATATTAGCAGGAGTGTTTGGTGGA	
*TNF*	Forward	GGTGCCTATGTCTCAGCCTCTT	139
	Reverse	GCCATAGAACTGATGAGAGGGAG	
*UCP-1*	Forward	ACTGCCACACCTCCAGTCATT	123
	Reverse	CTTTGCCTCACTCAGGATTGG	
*β-Actin*	Forward	CTAAGGCCAACCGTGAAAAG	104
	Reverse	ACCAGAGGCATACAGGGACA	

## Data Availability

Data associated with this study are available upon request.

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
