# Peer review of "Apolipoprotein CIII Reduction Protects White Adipose Tissues against Obesity-Induced Inflammation and Insulin Resistance in Mice"

_ijms, 2021, doi:10.3390/ijms23010062_

Round 1

Reviewer 1 Report

A VERY WELL WRITTEN AND INTERESTING PAPER REGARDING APOC 3 REDUCTION AND OBESITY-REDUCED INFLAMMATION IN MICE.

Very well written and interesting paper.

The fact that  apoC3 reduction can reduce obesity-induced inflammation and insulin resistance can lead to new drug development for prediabetes, DM2, chylomicronemia, atherosclerosis and metabolic syndrome patients (this comment should be added in the discussion part). ASOs development (volanesorsen) is currently a hot topic. Extensive animal, human and interventional data exist that apoC3 is a good pharmaceutical target for intervention to lower triglycerides but obesity-induced inflammation and insulin resistance have been less studied  A short comment regarding volanesorsen and human studies and what new knowledge adds this study would improve the paper.

Minor editing; Title should refer to the fact that these results were observed in mice  [“Apolipoprotein CIII reduction protects white adipose tissue against obesity-induced inflammation and insulin resistance in mice’’]

In addition, did the ASO-induced-triglyceride reduction was relevant to the observed obesity-induced inflammation and insulin resistance? Writers may add a comment.

Author Response

We appreciate the work of the referee and acknowledge the efforts reading the manuscript. We have gone through the comments of the reviewer and below you find the answers (written in blue). The changes in the revised manuscript are marked with tracking changes.

Reviewer 1

A VERY WELL WRITTEN AND INTERESTING PAPER REGARDING APOC 3 REDUCTION AND OBESITY-REDUCED INFLAMMATION IN MICE.

Very well written and interesting paper.

The fact that  apoC3 reduction can reduce obesity-induced inflammation and insulin resistance can lead to new drug development for prediabetes, DM2, chylomicronemia, atherosclerosis and metabolic syndrome patients (this comment should be added in the discussion part). ASOs development (volanesorsen) is currently a hot topic. Extensive animal, human and interventional data exist that apoC3 is a good pharmaceutical target for intervention to lower triglycerides but obesity-induced inflammation and insulin resistance have been less studied  A short comment regarding volanesorsen and human studies and what new knowledge adds this study would improve the paper.

Minor editing; Title should refer to the fact that these results were observed in mice  [“Apolipoprotein CIII reduction protects white adipose tissue against obesity-induced inflammation and insulin resistance in mice’’]

In addition, did the ASO-induced-triglyceride reduction was relevant to the observed obesity-induced inflammation and insulin resistance? Writers may add a comment.

  1. Question: A short comment regarding volanesorsen and human studies and what new knowledge adds this study would improve the paper.

Response: Volanesorsen is used in the clinic for patients with FCS and we agree that it is important to clarify what is new in this study and in the revised manuscript we have added a short section with a reference in the Discussion (lines 191-195 in the Discussion section; lines 402-405 in the References section).

  1. Question: Title should refer to the fact that these results were observed in mice  [“Apolipoprotein CIII reduction protects white adipose tissue against obesity-induced inflammation and insulin resistance in mice’’]

Response: Title has been changed accordingly (lines 4-5).

  1. Question: did the ASO-induced-triglyceride reduction was relevant to the observed obesity-induced inflammation and insulin resistance? Writers may add a comment.

Response: By lowering apoCIII you get like a chain-reaction. There is an increase in lipolysis and hepatic clearance of lipids, decrease in inflammation as apoCIII is known to be pro-inflammatory, and a normalization of insulin sensitivity. This has been described in detail in our publications in PNAS and Sci Adv (ref 9 and 10 in this paper), but to make it more clear we have, at the end of the Discussion (lines 236-237), added that besides the effects on metabolism there was a reduced expression of inflammatory cytokines.  

Reviewer 2 Report

The study focus on the role of Apo CIII modulation for  subcutaneous  and visceral  white adipose tissue using the mouse model.  The animals were  assigned either to  be fed a high-fat diet or kept on standard chow diet and  were treated with  Apo CIII antisense oligonucleotides

The manuscript is correctly structured, and the study seems well conducted. The results are relevant since the authors concluded that metabolic derangements in white adipose tissue can be prevented and reversed by lowering apoCIII.

Importantly, experimental data accordingly support conclusions. Nonetheless, some minor points should be addressed before publication.

The authors used non-parametric tests to assess the significance of differences between the two groups and parametric tests when analyzing a larger number of groups. What was the reason for such a decision?  It is also difficult to conclude when the authors drew conclusions on the basis of multiple comparisons and when the ANOVA test and appropriate pos hoc tests - a more detailed description of the figures could raise these doubts.

Author Response

We appreciate the work of the referee and acknowledge the efforts reading the manuscript. We have gone through the comments of the reviewer and below you find the answers (written in blue). The changes in the revised manuscript are marked with tracking changes.

Reviewer 2

The study focus on the role of Apo CIII modulation for  subcutaneous  and visceral  white adipose tissue using the mouse model.  The animals were  assigned either to  be fed a high-fat diet or kept on standard chow diet and  were treated with  Apo CIII antisense oligonucleotides

The manuscript is correctly structured, and the study seems well conducted. The results are relevant since the authors concluded that metabolic derangements in white adipose tissue can be prevented and reversed by lowering apoCIII.

Importantly, experimental data accordingly support conclusions. Nonetheless, some minor points should be addressed before publication.

The authors used non-parametric tests to assess the significance of differences between the two groups and parametric tests when analyzing a larger number of groups. What was the reason for such a decision?  It is also difficult to conclude when the authors drew conclusions on the basis of multiple comparisons and when the ANOVA test and appropriate pos hoc tests - a more detailed description of the figures could raise these doubts.

Response: We thank the reviewer for the comments and for raising the point regarding the statistical analyses. We apologize for the unclarity. In the current work the conclusions were drawn based on the non-parametric two-tailed Mann-Whitney U t-tests comparing Scr14w versus CD14w and ASO14w versus Scr14w, for the prevention study; and HFD+Scr versus CD+Sal and HFD+ASO verus HFD+Scr for the reversibility study. Hence, we have removed 1-ANOVA followed by Tukey’s post-hoc test as these tests were not performed in this study (lines 331-333), but we have used them in other studies and  by mistake they were included in the section  “Statistical analysis”.